# IMPROVING MLP MODULE IN VISION TRANSFORMER

## ABSTRACT

Transformer models have been gaining substantial interest in the field of computer vision tasks nowadays. Although a vision transformer contains two important components which are self-attention module and multi-layer perceptron (MLP) module, the majority of research tends to concentrate on modifying the former while leaving the latter in its original form. In this paper, we focus on improving the MLP module within the vision transformer. Through theoretical analysis, we demonstrate that the effect of the MLP module primarily lies in providing non-linearity, whose degree corresponds to the hidden dimensions. Thus, the computational cost of the MLP module can be reduced by enhancing the degree of non-linearity in the nonlinear function. Leveraging this insight, we propose an improved MLP (IMLP) module for vision transformers which involves the usage of the arbitrary GeLU (AGeLU) function and integrating multiple instances of it to augment non-linearity so that the number of hidden dimensions can be effectively reduced. Besides, a spatial enhancement part is involved to further enrich the non-linearity in the proposed IMLP module. Experimental results show that we can apply our method to a wide range of state-of-the-art vision transformer models irrespective of how they modify their self-attention part and the overall architecture, and reduce FLOPs and parameters without compromising classification accuracy on the ImageNet dataset.

## 1 INTRODUCTION

Transformer models with self-attention operation have been applied to the field of computer vision and achieve impressive results on many tasks such as image classification (Dosovitskiy et al., 2021; Touvron et al., 2021), object detection (Fang et al., 2021), semantic segmentation (Strudel et al., 2021) and video analysis (Neimark et al., 2021) nowadays. Compared to convolutional neural networks (CNNs), transformer models have less inductive bias due to the low-pass filter property of the self-attention (Park & Kim, 2022) and have the capability to utilize more training data to enhance generalization ability. However, when given a limited amount of training data such as ImageNet-1k, the original Vision Transformer (ViT) model (Dosovitskiy et al., 2021) cannot perform on par with state-of-the-art CNN models, making it difficult to apply ViT to complicated vision tasks in reality.

The modification of the vanilla ViT model primarily lies in two different parts. The first one is to change the basic architecture of ViT. Hierarchical ViTs (Heo et al., 2021; Liu et al., 2021) leverage the advantage of hierarchical architecture of CNNs and reduce the spatial size as well as expand the channel dimensions multiple times with the help of pooling layers. A convolution stem with multiple convolutional layers is introduced in He et al. (2019) to replace the non-overlapping patch embedding operation. The second one is to modify the self-attention module in ViT. Local-enhanced vision transformers (Huang et al., 2021; Wu et al., 2022) constrain the range of attention and generate patches within a local region, and then facilitate interactions between these patches to extract and interpret global information. Efficient self-attention operations reduce computational complexity of previous self-attention operation from $\mathcal{O}(n^2)$ to $\mathcal{O}(n)$ (Wang et al., 2020) or $\mathcal{O}(nlog(n))$ (Kitaev et al., 2020).

Although a substantial number of works concentrate on studying the variations of vision transformers, very few of them pay attention to modifying the multi-layer perceptron (MLP) module. CMT (Guo et al., 2022) uses an inverted residual feed-forward network to replace the original MLP module, CoAtNet (Dai et al., 2021) uses MBConv blocks (Sandler et al., 2018) to replace some of the ViT blocks in its network architecture. However, there are multiple modifications in their archi-

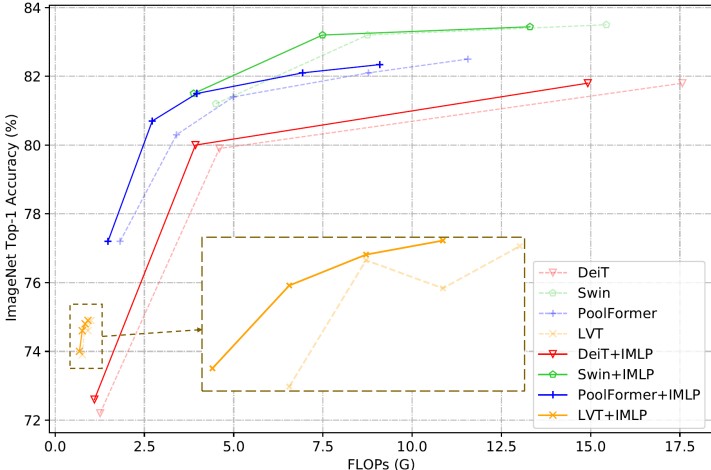

Figure 1: Top-1 classification accuracy versus FLOPs for different models on ImageNet-1k dataset. Our IMLP module can reduce FLOPs without sacrificing classification performance on different baseline vision transformer models.

tectures and the effectiveness of modifying MLP module remains unclear. Furthermore, there is a lack of theoretical analysis explaining why these changes are effective.

In this paper, we first give a thorough analysis of the MLP module in the vision transformer and show that the effect of the MLP module primarily lies in providing non-linearity whose degree corresponds to the hidden dimensions. Then, a rather intuitive idea comes that if we can enhance the degree of non-linearity in the nonlinear function, we could potentially decrease the hidden dimensions of the MLP module, thereby reducing the computational cost. Based on this thought, we introduce the arbitrary GeLU (AGeLU) function which is easy to combine to generate stronger non-linearity. Besides, a spatial-wise enhancement part is added to further enrich the non-linearity of the module. By combining them together, we introduce our improved MLP (IMLP) module for vision transformer. We conduct several experiments on different popular vision transformer models with various designs of the whole architecture and self-attention module including DeiT, Swin, PoolFormer, LVT, *etc.*, by replacing their original MLP module into the proposed IMLP module. Results on ImageNet-1k dataset show that we can effectively reduce FLOPs and parameters without sacrificing the classification accuracy as shown in Fig. 1 and the experiment section.

## 2 RELATED WORKS

Vision transformer (ViT) was first introduced by Dosovitskiy et al. (2021) to extend the transformer architecture to vision tasks. Since then, researches have focused on improving the performance of vanilla ViT. For example, DeiT (Touvron et al., 2021) leveraged the knowledge distillation method and introduced a series of new training techniques to enhance the classification performance on ImageNet-1k. Swin (Liu et al., 2021) utilized a hierarchical architecture and adopted a local self-attention mechanism to reduce computational complexity while using shift operation to add interaction across different sub-windows. PoolFormer (Yu et al., 2022) argued that the whole architecture of ViT was more important than the self-attention operation and replaced the multi-head self-attention (MHSA) modules with pooling operations.

Methods above focus on modifying the training strategy, the whole architecture of ViT and the MHSA module. Very little research studied the MLP module in ViT. CMT (Guo et al., 2022) and PVTv2 (Wang et al., 2022) introduced ViT models with several modifications, and one of them was to use the inverted residual feed-forward network to replace the MLP module. CoAtNet (Dai et al., 2021) found that vertically stacking convolution layers and attention layers was surprisingly effective and replaced some of the ViT blocks with MBConv blocks through neural architecture search. These studies generated new ViT models with various modifications to the fundamental architecture. However, the impact of altering the MLP module remains uncertain.

Figure 2: An intuitive illustration of the corollary that the MLP module is a non-linearity generator. We use $\phi(\cdot) = \text{ReLU}(\cdot)$ in this figure for simplicity. Other formats of nonlinear function can also be used here to derive the same conclusion.

## 3  MLP MODULE IS A NON-LINEARITY GENERATOR

Considering an input matrix $\boldsymbol{X} \in \mathbb{R}^{N \times C}$ in which $N$ is the number of patches and $C$ is the dimension of each patch, the output of the MLP module can be calculated as:

$$\boldsymbol{Y} = \text{MLP}(\boldsymbol{X}) = \phi(\boldsymbol{X}\boldsymbol{W}^a)\boldsymbol{W}^b, \tag{1}$$

where $\boldsymbol{W}^a = \{w_{ij}^a\} \in \mathbb{R}^{C \times C'}$ and $\boldsymbol{W}^b = \{w_{ij}^b\} \in \mathbb{R}^{C' \times C}$ are weight matrices of two FC layers, $C'$ controls the number of hidden dimensions, and $\phi(\cdot)$ represents the non-linear function. $C' = 4C$ and $\phi(\cdot) = \text{GeLU}(\cdot)$ are used in the original ViT model.

Without loss of generality, we assume $N = 1$ and the input matrix $\boldsymbol{X}$ degrades into an input vector $\boldsymbol{x} \in \mathbb{R}^C$. Then, we can represents Eq. 1 in its element-wise form:

$$\boldsymbol{x}\boldsymbol{W}^a = \left(\sum_{i=1}^{C} w_{ic'}^a x_i\right)_{c'=1}^{C'}, \quad \phi(\boldsymbol{x}\boldsymbol{W}^a) = \left(\phi(\sum_{i=1}^{C} w_{ic'}^a x_i)\right)_{c'=1}^{C'},$$

$$\boldsymbol{y} = \phi(\boldsymbol{x}\boldsymbol{W}^a)\boldsymbol{W}^b = \left(\sum_{j=1}^{C'} w_{jc}^b \phi(\sum_{i=1}^{C} w_{ij}^a x_i)\right)_{c=1}^{C} = \left(\sum_{j=1}^{C'} w_{jc}^b \phi(m_{cj} x_c + n_{cj})\right)_{c=1}^{C}, \tag{2}$$

in which $m_{cj} = w_{cj}^a$ and $n_{cj} = f(x_1, \cdots, x_{c-1}, x_{c+1}, \cdots, x_C) = \sum_{i=1, i \neq c}^{C} w_{ij}^a x_i$. Given Eq. 2, we can derive the following corollary:

**Corollary 1** *Given an input vector $\boldsymbol{x} \in \mathbb{R}^C$, the output of the MLP module in Eq. 1 is denoted as $\boldsymbol{y} \in \mathbb{R}^C$. Then:*

*(1) Each element $y_c$ in $\boldsymbol{y}$ is the linear combination of $C'$ different nonlinear functions to the input element $x_c$.*

*(2) Distinct scales and biases are applied to different input elements $x_c$ before passing through the nonlinear function $\phi(\cdot)$.*

*(3) The scale is a learnable weight independent to the input element $x_c$, while the bias is dependent to all other input elements in $\boldsymbol{x}$.*

The above conclusion brings to light that the MLP module in the vision transformer is no more than a non-linearity generator with a nonlinear degree of $C'$, as intuitively shown in Fig. 2.

## 4  METHOD

### 4.1  A MORE POWERFUL NONLINEAR FUNCTION

Based on the corollary in the previous section, a straightforward way is to use the combination of $C'$ different nonlinear functions to replace the original MLP module. However, the bias which depends on the input elements makes it challenging to attain a comparable degree of non-linearity by merely combining multiple nonlinear functions, and the classification performance does not match that of using the original MLP module (as shown in Tab. 4).

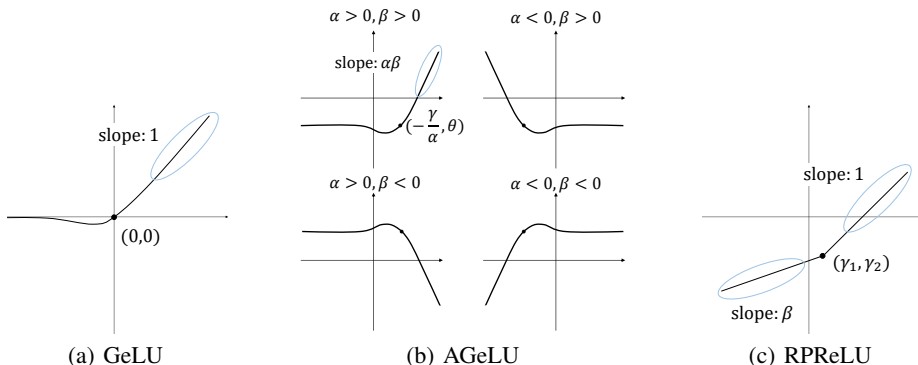

Figure 3: The comparison among the shapes of GeLU, AGeLU and RPReLU.

In the following paragraph, we first introduce the arbitrary nonlinear function which is flexible and easy to be concatenated together to form a more powerful nonlinear function. Subsequently, we demonstrate that the hidden dimension of the MLP module can be effectively reduced with this enhanced nonlinear function.

**Arbitrary nonlinear function.** Arbitrary nonlinear function is defined as

$$\phi'(x) = \beta\phi(\alpha x + \gamma) + \theta, \tag{3}$$

in which $x$ is the input of the arbitrary nonlinear function, $\alpha$ and $\beta$ are learnable coefficients before and after applying the basic nonlinear function $\phi(\cdot)$, and $\gamma$ and $\theta$ are learnable biases. The inspiration for introducing arbitrary nonlinear function arises from Eq. 2 where distinct weights and biases are employed to each element $x_c$ before and after applying the basic nonlinear function. Since GeLU is used as a basic nonlinear function in ViT, we introduce the arbitrary GeLU (AGeLU) to our model:

$$\text{AGeLU}(x) = \beta\text{GeLU}(\alpha x + \gamma) + \theta. \tag{4}$$

AGeLU is more flexible than other modified nonlinear functions such as the RPReLU function proposed in ReActNet (Liu et al., 2020). The latter can only adjust the position of the turning point compared to PReLU, while AGeLU can also provide a learnable slope of the function and switch the whole shape by using different positive and negative coefficients $\alpha$ and $\beta$. Fig. 3 gives a comparison among the shapes of GeLU, AGeLU, and RPReLU. Note that other basic activation functions such as ReLU, PReLU, *etc*. can be extended using the same way as AGeLU to form AReLU and APReLU.

**Reducing the hidden dimension of MLP module with powerful nonlinear function.** Rather than using the original MLP module introduced in Eq. 1, we propose our AMLP module that integrates two AGeLU functions and forms a powerful nonlinear function to replace the original GeLU and halve the hidden dimension of the module. Specifically, we have:

$$\boldsymbol{Y}' = \text{AMLP}(\boldsymbol{X}) = \text{concat}(\text{AGeLU}(\boldsymbol{X}\boldsymbol{W}^d), \text{AGeLU}'(\boldsymbol{X}\boldsymbol{W}^d))\boldsymbol{W}^e, \tag{5}$$

where $\boldsymbol{W}^d = \{w_{ij}^d\} \in \mathbb{R}^{C \times \frac{C'}{2}}$ and $\boldsymbol{W}^e = \{w_{ij}^e\} \in \mathbb{R}^{C' \times C}$ are weight matrices of two FC layers, and $\text{AGeLU}(\cdot)$ and $\text{AGeLU}'(\cdot)$ are two nonlinear functions proposed in Eq. 4 with different parameters. With this simple modification, the first FC layer has half the output channels compared to the original MLP module, and can effectively reduce the FLOPs and parameters in the vision transformer model. In the following section, we show that the proposed AMLP module can also be treated as the linear combination of $C'$ different nonlinear functions.

We can degrade the input matrix $\boldsymbol{X}$ into an input vector $\boldsymbol{x} \in \mathbb{R}^C$, and represent Eq. 5 in its element-wise form:

$$\boldsymbol{t}_0 = \boldsymbol{x}\boldsymbol{W}^d = \left(\sum_{i=1}^{C} w_{ic'}^d x_i\right)_{c'=1}^{\frac{C'}{2}},$$

$$\boldsymbol{t}_1 = \text{AGeLU}(\boldsymbol{t}_0) = \left(\beta_{c'}\text{GeLU}(\alpha_{c'}\sum_{i=1}^{C} w_{ic'}^d x_i + \gamma_{c'}) + \theta_{c'}\right)_{c'=1}^{\frac{C'}{2}},$$

$$\boldsymbol{t}'_1 = \text{AGeLU}'(\boldsymbol{t}_0) = \left( \beta'_{c'} \text{GeLU}(\alpha'_{c'} \sum_{i=1}^{C} w^d_{ic'} x_i + \gamma'_{c'}) + \theta'_{c'} \right)_{c'=1}^{\frac{C'}{2}},$$

$$\boldsymbol{t}_2 = \text{concat}(\boldsymbol{t}_1, \boldsymbol{t}'_1) = \left( \beta_{c'} \text{GeLU}(\alpha_{c'} \sum_{i=1}^{C} w^d_{i,f(c')} x_i + \gamma_{c'}) + \theta_{c'} \right)_{c'=1}^{C'},$$

$$\boldsymbol{y}' = \boldsymbol{t}_2 \boldsymbol{W}^e = \left( \sum_{j=1}^{C'} w^e_{jc} \cdot [\beta_j \text{GeLU}(\alpha_j \sum_{i=1}^{C} w^d_{i,f(j)} x_i + \gamma_j) + \theta_j] \right)_{c=1}^{C},$$

$$= \left( \sum_{j=1}^{C'} w^{'e}_{jc} \text{GeLU}(m'_{cj} x_c + n'_{cj}) + \theta_j \right)_{c=1}^{C}, \tag{6}$$

where in the fourth line, we define $\alpha'_1, \cdots, \alpha'_{\frac{C'}{2}} \triangleq \alpha_{\frac{C'}{2}+1}, \cdots, \alpha_{C'}$ (the same to $\beta'$, $\gamma'$ and $\theta'$), and $f(x) = x - \frac{C'}{2} \cdot \mathbb{1}_{x > \frac{C'}{2}}$ in which $\mathbb{1}$ is the indicator function. $w^{'e}_{jc} = w^e_{jc} \cdot \beta_j$, $m'_{cj} = w^d_{c,f(j)}$ and $n'_{cj} = \text{func}(x_1, \cdot, \cdot, \cdot, x_{c-1}, x_{c+1}, \cdot, \cdot, \cdot, x_C) = \sum_{i=1,i\neq c}^{C} w^d_{i,f(j)} x_i + \gamma_j$.

Note that it is almost the same compared to the original MLP module (Eq. 2), the proposed AMLP module (Eq. 6) is also a generator that generates the same degree of non-linearity. Each element $y'_c$ in $\boldsymbol{y}'$ can also be treated as a linear combination of $C'$ different nonlinear functions to the input element $x_c$, each with distinct scales and biases. Each scale is a learnable weight while each bias is dependent on other input elements.

## 4.2 THEORETICAL ANALYSIS

In this section, we analyze the Lipschitz constant of the proposed AMLP module. Note that the Lipschitz constant serves as a metric for assessing the network's stability by bounding the rate of output change in response to input perturbations, while also highlighting the network's susceptibility to adversarial attacks. Thus, it is beneficial to study the Lipschitz constant that contributes to improving the reliability of our module.

Firstly, we give the definition of a Lipschitz constant:

**Definition 1** *A function $f : \mathbb{R}^n \to \mathbb{R}^m$ is Lipschitz continuous if there exists a non-negative constant $L$ such that*

$$||f(x) - f(y)||_2 \leq L||x - y||_2 \quad \text{for all } x, y \in \mathbb{R}^n, \tag{7}$$

*among which the smallest $L$ is called the Lipschitz constant of function $f$.*

In the following paragraph, we present a lemma to describe the conceptualization of nonlinear activation functions, and then use a theorem to derive the bound on the Lipschitz constant of our proposed AMLP module.

**Lemma 1** *(Fazlyab et al., 2019) Suppose $\varphi : \mathbb{R} \to \mathbb{R}$ is slope-restricted on $[p, q]$. Define the set*

$$\mathcal{T}_n = \{T \in \mathbb{S}^n | T = \sum_{i=1}^{n} \lambda_{ii} e_i e_i^\top, \lambda_{ii} \geq 0\}. \tag{8}$$

*Then for any $T \in \mathcal{T}_n$ the vector-valued function $\phi(x) = [\varphi(x_1), \cdots, \varphi(x_n)]^\top : \mathbb{R}^n \to \mathbb{R}^n$ satisfies*

$$\begin{bmatrix} x - y \\ \phi(x) - \phi(y) \end{bmatrix}^\top \begin{bmatrix} -2pqT & (p+q)T \\ (p+q)T & -2T \end{bmatrix} \begin{bmatrix} x - y \\ \phi(x) - \phi(y) \end{bmatrix} \geq 0 \quad \text{for all } x, y \in \mathbb{R}^n.$$

It is easy to prove that our proposed AGeLU activation function satisfies the slope-restricted condition when the parameters $\alpha$ and $\beta$ in Eq. 4 are finite. The matrix $T$ is used for deriving the Lipschitz bound of the AMLP module in the following theorem.

**Theorem 1** *Given the AMLP module described by $f(x) = W^1\text{concat}(\phi_1(W^0 x + b^0), \phi_2(W^0 x + b^0)) + b^1$. Suppose $\phi_i(x) : \mathbb{R}^n \to \mathbb{R}^n = [\varphi_i(x_1), \cdots, \varphi_i(x_n)]$, where $\varphi_i$ is slope-restricted on $[p_i, q_i], i \in \{1, 2\}$. Define $\mathcal{T}_n$ as in Eq. 8. Suppose there exists $\rho_1, \rho_2 > 0$ such that the matrix inequalities*

$$M(\rho_i, T) := \begin{bmatrix} -2p_i q_i W^{0\top} T W^0 - \rho_i I_{n_0} & (p_i + q_i) W^{0\top} T \\ (p_i + q_i) T W^0 & -2T + W^{1i\top} W^{1i} \end{bmatrix} \preceq 0, \quad i \in \{1, 2\}, \quad (9)$$

*holds for some $T \in \mathcal{T}_n$, where $W^1 = [W^{11} \ W^{12}]$. Then $||f(x) - f(y)||_2 \le (\sqrt{\rho_1} + \sqrt{\rho_2})||x - y||_2$ for all $x, y \in \mathbb{R}^{n_0}$.*

Theorem 1 gives an upper bound of $L(f) = \sqrt{\rho_1} + \sqrt{\rho_2}$ on the Lipschitz constant of the AMLP module $f(x) = W^1\text{concat}(\phi_1(W^0 x + b^0), \phi_2(W^0 x + b^0)) + b^1$. The above equation can be treated as a semi-definite program (SDP) which can be solved numerically to derive its global minimum. The proof of Theorem 1 is in the Appendix A.1.

### 4.3 ENHANCING NON-LINEARITY WITH SPATIAL INFORMATION

Although the AMLP module generates a same degree of non-linearity compared to the original MLP module, we notice that the degree of freedom of $\{w_{i,f(j)}^d\}_{j=1}^{C'}$ in Eq. 6 are halved compared to the original $\{w_{ij}^a\}_{j=1}^{C'}$ in Eq. 2. It is similar to the model quantization methods that halve the number of bits used for weights and activation and may degrade the performance. Thus, a straight-forward way is to treat the original vision transformer model as the teacher model and utilize the knowledge distillation (KD) method (Hinton et al., 2015) to distill the output of the AMLP module. However, as the size of the model grows larger, this approach becomes infeasible due to insufficient GPU memory for the KD training process.

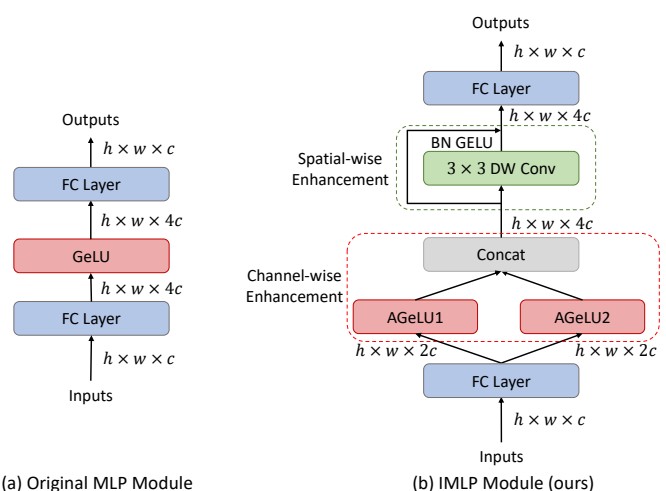

Figure 4: The architecture of (a) the original MLP module and (b) the proposed IMLP module. The channel-wise enhancement part includes the AGeLU function and concatenation operation. The spatial-wise enhancement part includes a depthwise block.

In the previous section, we extend the non-linearity of the MLP module through the channel dimension. Therefore, in this section we further enhance non-linearity with spatial information. Many previous studies use convolution operation in vision transformers. For example, CMT (Guo et al., 2022) uses inverted residual FFN in the network, and CoAtNet (Dai et al., 2021) replaces some of the attention blocks with inverted bottlenecks. However, they do not mention the relationship between these blocks and the extension of non-linearity. VanillaNet (Chen et al., 2023) proposes series informed activation function to enrich the approximation ability which is formulated as:

$$\phi_s(x_{h,w,c}) = \sum_{i,j \in \{-n,n\}} a_{i,j,c} \phi(x_{i+h,j+w,c} + b_c), \quad (10)$$

where $\phi(\cdot)$ is the activation function. We found that this is equal to going through the non-linear function followed by a $n \times n$ depthwise convolution (DW Conv), which means that DW Conv after the non-linear function utilizes the spatial information and enhances non-linearity by learning global information from its neighbors. Thus, we modify our AMLP module by introducing a DW Block (DW Conv with BN and GeLU) after AGeLU, and form the final improved MLP (IMLP)

module as shown in Fig. 4. The IMLP module has two main differences compared to the original MLP module. The first is the channel-wise enhancement part that includes the AGeLU function and concatenation operation proposed in section 4.1 to extend non-linearity through channel dimension. The second is the spatial-wise enhancement part with a DW Block to enhance non-linearity with spatial information. A $3 \times 3$ DW Conv is used as default unless specified in the experiments.

## 5 EXPERIMENTS

In this section, we conduct experiments on the ImageNet-1k dataset for image classification and then ablate different parts of IMLP through ablation studies. Experiments on object detection and semantic segmentation are shown in the Appendix A.2 and A.3.

### 5.1 IMAGE CLASSIFICATION ON IMAGENET-1K

We empirically verify the effectiveness of the proposed IMLP module on the ImageNet-1k dataset which contains 1.28M training images from 1000 different classes and 50K validation images.

**Implementation details.** We treat our IMLP module as a plug-in and replacement module that is used to replace the original MLP in different vision transformers. Thus the training strategies are exactly the same as the original methods. There are two hyper-parameters that can be tuned in the IMLP module. The first is the expansion ratio $r$ of the first FC layer, and the second is the kernel size $n$ of the depthwise convolution operation in the spatial-wise enhancement part. We use $r = 2$ and $n = 3$ in the following experiments if not specified.

**Baseline models.** We select several widely used state-of-the-art vision transformer models as our baseline models, including DeiT (Touvron et al., 2021), Swin (Liu et al., 2021), PoolFormer (Yu et al., 2022) and portable vision transformer such as LVT (Yang et al., 2022).

**Experimental results.** We replace all the MLP modules in each baseline method with the proposed IMLP module. The experimental results are shown in Tab. 1. We can see that almost all the models can reduce over 10% FLOPs and parameters without loss of classification accuracy. For example, we can reduce the parameter count of the DeiT-Ti model by 12.6% and FLOPs by 12.7% while increasing the top-1 accuracy by 0.4%. As the model becomes larger, the amount of parameter/FLOPs reduction also increases as the proportion of the MLP module in the computation grows. Similar results can be seen in other baseline models. PoolFormer models exhibit higher FLOPs and parameter reduction (over 20%) since most of their calculations come from the MLP module.

### 5.2 ABLATION STUDIES

In this section, we ablate various design choices for each part of the IMLP module to empirically verify the effectiveness of the proposed method.

**Effect of channel/spatial-wise enhancement part.** In Tab. 2 we separately use channel-wise and spatial-wise enhancement parts in the IMLP module. When using channel-wise enhancement alone, we also verify the effectiveness of the KD method. When using spatial-wise enhancement alone, we replace the channel-wise enhancement part with the original GeLU activation function. We can see that channel-wise enhancement can reduce the FLOPs and parameters of the model but there is a performance degradation compared to the baseline. Using the KD method can make up for the gap but the GPU memory usage during training will be increased significantly, thus is abandoned in our method. Combining the channel-wise and spatial-wise enhancement brings about a smaller model with better classification accuracy.

**Effect of AGeLU function.** Note that we use two AGeLU functions and a concatenation operation in the channel-wise enhancement part. In the following experiment, we compare the classification performance of using AGeLU and GeLU functions with and without concatenation operation. The first line is the baseline DeiT-Ti model, and the second line changes GeLU in MLP modules to AGeLU. Line 4 is the proposed method, and line 3 changes the proposed AGeLU to GeLU. We observe from Tab. 3 that when setting the expansion ratio $r = 4$ without concatenation, the proposed AGeLU function does not demonstrate superiority over the original GeLU function. However, when using $r = 2$ with concatenation, the AGeLU function exhibits improved performance. This is

Table 1: Image classification results on ImageNet-1k datasets. Several widely used state-of-the-art vision transformer models are used as the baseline models, and the original MLP modules in them are replaced with the proposed IMLP module. '*' indicates that we use $n = 5$ for the depthwise convolution operation in the spatial-wise enhancement part.

| Methods | Architecture | Parameters (M) | FLOPs (G) | Top-1 Accuracy (%) |
|---|---|---|---|---|
| DeiT | DeiT-Ti | 5.72 | 1.26 | 72.2 |
| | + IMLP | 5.00 (-12.6%) | 1.10 (-12.7%) | 72.6 |
| | DeiT-S | 22.05 | 4.60 | 79.9 |
| | + IMLP | 18.84 (-14.6%) | 3.93 (-14.6%) | 80.0 |
| | DeiT-B | 86.57 | 17.57 | 81.8 |
| | + IMLP* | 73.66 (-14.9%) | 14.92 (-15.1%) | 81.8 |
| Swin | Swin-Ti | 28.29 | 4.50 | 81.2 |
| | + IMLP | 24.29 (-14.1%) | 3.88 (-13.8%) | 81.5 |
| | Swin-S | 49.61 | 8.75 | 83.2 |
| | + IMLP | 42.40 (-14.5%) | 7.49 (-14.4%) | 83.2 |
| | Swin-B | 87.77 | 15.44 | 83.5 |
| | + IMLP* | 75.45 (-14.0%) | 13.34 (-13.6%) | 83.4 |
| PoolFormer | PoolFormer-S12 | 11.92 | 1.82 | 77.2 |
| | + IMLP | 9.80 (-17.8%) | 1.48 (-18.7%) | 77.2 |
| | PoolFormer-S24 | 21.39 | 3.40 | 80.3 |
| | + IMLP | 17.15 (-19.8%) | 2.72 (-20.0%) | 80.7 |
| | PoolFormer-S36 | 30.86 | 4.99 | 81.4 |
| | + IMLP | 24.50 (-20.6%) | 3.97 (-20.4%) | 81.5 |
| | PoolFormer-M36 | 56.17 | 8.78 | 82.1 |
| | + IMLP | 44.19 (-21.3%) | 6.93 (-21.1%) | 82.1 |
| | PoolFormer-M48 | 73.47 | 11.56 | 82.5 |
| | + IMLP* | 58.62 (-20.2%) | 9.46 (-18.2%) | 82.3 |
| Portable ViT | LVT-R1 | 5.52 | 0.76 | 73.9 |
| | + IMLP* | 4.98 (-9.8%) | 0.68 (-10.5%) | 74.0 |
| | LVT-R2 | 5.52 | 0.84 | 74.8 |
| | + IMLP* | 4.98 (-9.8%) | 0.76 (-9.5%) | 74.6 |
| | LVT-R3 | 5.52 | 0.92 | 74.6 |
| | + IMLP* | 4.98 (-9.8%) | 0.84 (-8.7%) | 74.8 |
| | LVT-R4 | 5.52 | 1.00 | 74.9 |
| | + IMLP* | 4.98 (-9.8%) | 0.92 (-8.0%) | 74.9 |

Table 2: Ablation study on channel/spatial-wise enhancement part. The experiments are conducted using the DeiT-Ti model on the ImageNet dataset.

| Methods | Parameters (M) | FLOPs (G) | Top-1 Accuracy (%) |
|---|---|---|---|
| DeiT-Ti | 5.72 | 1.26 | 72.2 |
| w/ channel | 4.89 | 1.08 | 70.5 |
| w/ channel + KD | 4.89 | 1.08 | 72.0 |
| w/ spatial | 5.83 | 1.28 | 72.8 |
| w/ channel & spatial | 5.00 | 1.10 | 72.6 |

because in the former setting the two activation functions have the same degree of non-linearity, while in the latter the two GeLU functions with the same scale and bias cause a simple replication along the channel dimension and lead to a degradation of the performance and the two AGeLU functions do not decrease the degree of non-linearity (according to the analysis in Sec. 3).

Table 3: Ablation study on whether or not using AGeLU function and the concatenation operation in channel-wise enhancement part. The experiments are conducted using the DeiT-Ti model on the ImageNet dataset.

| Methods | ratio | Top-1 Acc (%) |
|---|---|---|
| GeLU (baseline) | 4 | 72.2 |
| AGeLU | 4 | 72.1 |
| GeLU+concat | 2 | 72.3 |
| AGeLU+concat (ours) | 2 | 72.6 |

Table 4: Using the addition of $4C$ number of nonlinear functions to replace the original MLP module. GeLU and AGeLU are used as the basic nonlinear functions. The experiments are conducted based on the DeiT-Ti model.

| Methods | Top-1 Acc (%) |
|---|---|
| DeiT-Ti | 72.2 |
| GeLU | 50.4 |
| AGeLU | 53.3 |

Table 5: Using different kernel sizes $n$ for depthwise convolution in the spatial-wise enhancement part. The experiments are conducted using the DeiT-Ti model on the ImageNet dataset.

| $n$ | Parameters (M) | FLOPs (G) | Top-1 Accuracy (%) |
|---|---|---|---|
| 1 | 4.92 | 1.08 | 72.0 |
| 3 | 5.00 | 1.10 | 72.6 |
| 5 | 5.15 | 1.13 | 72.8 |
| 7 | 5.37 | 1.17 | 72.9 |

**Effect of directly combining multiple nonlinear functions.** In Sec. 4.1 we analyze that the straightforward way of directly using $4C$ different nonlinear functions to replace the original MLP module is challenging to attain a comparable degree of non-linearity. Here we use GeLU and AGeLU as the basic nonlinear functions. In Tab. 4 we empirically show the results of adding $4C$ number of nonlinear functions (*e.g.* $y = \text{AGELU}_1(x) + \cdots + \text{AGELU}_{4C}(x)$). We can see that none of these variants is comparable to the classification performance of the baseline with the MLP module, since according to Corollary 1 the biases of these nonlinear functions should be different and are dependent on all other input elements which is hard to apply in reality and is the main reason that causes the performance degradation. Experiments are conducted using the DeiT-Ti model on the ImageNet dataset.

**Effect of using different kernel-size.** Finally, we use different kernel size for depthwise convolution in the spatial-wise enhancement part to explore the relationship between the classification performance and the amount of spatial information used to enhance the non-linearity. In Tab. 5, we can see that as the kernel size $n$ increases, the classification performances are getting better and better with a little increased FLOPs and parameters. The benefit is obvious from $n = 1$ to $n = 3$ since global information from the neighbors are used. The profit becomes marginal as the kernel size continues to grow.

## 6 CONCLUSION

In this paper, we thoroughly analyze the effect of the MLP module in the vision transformer and show that the original MLP module is no more than a non-linearity generator whose nonlinear degree corresponds to the number of hidden dimensions. Based on this observation, we propose a flexible activation function AGeLU and combine multiple of them to form a more powerful nonlinear function that extends non-linearity through channel dimension. Furthermore, we enhance non-linearity with spatial information using depthwise block. With the above modification, we can use fewer hidden dimensions which reduces the FLOPs and parameters of the model without loss of classification performance. We also give a theoretical analysis of the Lipschitz bound of the proposed module by which the stability of the network can be measured. We conduct experiments on several state-of-the-art vision transformer models using the benchmark dataset ImageNet-1k by replacing the original MLP module with the proposed IMLP module, and the results demonstrate the effectiveness of our method.

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

## A APPENDIX

### A.1 PROOF OF THEOREM 1

Given

$$f(x) = W^1 \text{concat}(\phi_1(W^0 x + b^0), \phi_2(W^0 x + b^0)) + b^1, \qquad (11)$$

it is easy to rewrite the function as

$$
\begin{aligned}
f(x) &= g(x) + h(x) \\
&= \left( W^{11}\phi_1(W^0 x + b^0) + b^{11} \right) + \left( W^{12}\phi_2(W^0 x + b^0) + b^{12} \right),
\end{aligned} \qquad (12)
$$

in which $W^1 = [W^{11} \ W^{12}]$ and $b^1 = [b^{11} \ b^{12}]$. Thus, the function $f(x)$ can be divided into two parts $g(x)$ and $h(x)$. In the following analysis, we give the proof of Lipschitz bound on $g(x)$, and the bound on $h(x)$ can be derived in the same way.

Define $x^1 = \phi_1(W^0 x + b^0) \in \mathbb{R}^n$ and $y^1 = \phi_1(W^0 y + b^0) \in \mathbb{R}^n$ for two arbitrary inputs $x, y \in \mathbb{R}^{n_0}$. Using the conclusion in Lemma 1, we have:

$$
\begin{bmatrix} (W^0 x + b^0) - (W^0 y + b^0) \\ x^1 - y^1 \end{bmatrix}^\top
\begin{bmatrix} -2p_1 q_1 T & (p_1 + q_1)T \\ (p_1 + q_1)T & -2T \end{bmatrix}
\begin{bmatrix} (W^0 x + b^0) - (W^0 y + b^0) \\ x^1 - y^1 \end{bmatrix} \geq 0,
$$

where $T \in \mathcal{T}_n$ (Eq. 8). The above inequality can be rewritten as:

$$
\begin{bmatrix} x - y \\ x^1 - y^1 \end{bmatrix}^\top
\begin{bmatrix} -2p_1 q_1 W^{0\top} T W^0 & (p_1 + q_1)W^{0\top} T \\ (p_1 + q_1)T W^0 & -2T \end{bmatrix}
\begin{bmatrix} x - y \\ x^1 - y^1 \end{bmatrix} \geq 0, \qquad (13)
$$

By left and right multiply $M(\rho_1, T)$ in Eq. 9 by $[(x - y)^\top \ (x^1 - y^1)^\top]$ and $[(x - y)^\top \ (x^1 - y^1)^\top]^\top$ respectively, we have:

$$
\begin{aligned}
&\begin{bmatrix} x - y \\ x^1 - y^1 \end{bmatrix}^\top
\begin{bmatrix} -2p_1 q_1 W^{0\top} T W^0 & (p_1 + q_1)W^{0\top} T \\ (p_1 + q_1)T W^0 & -2T \end{bmatrix}
\begin{bmatrix} x - y \\ x^1 - y^1 \end{bmatrix} \\
&\qquad\qquad \leq
\begin{bmatrix} x - y \\ x^1 - y^1 \end{bmatrix}^\top
\begin{bmatrix} \rho_1 I_{n_0} & 0 \\ 0 & -W^{11\top} W^{11} \end{bmatrix}
\begin{bmatrix} x - y \\ x^1 - y^1 \end{bmatrix}.
\end{aligned} \qquad (14)
$$

Combining Eq. 13 and Eq. 14, we have:

$$0 \leq \begin{bmatrix} x - y \\ x^1 - y^1 \end{bmatrix}^\top \begin{bmatrix} \rho_1 I_{n_0} & 0 \\ 0 & -W^{11^\top} W^{11} \end{bmatrix} \begin{bmatrix} x - y \\ x^1 - y^1 \end{bmatrix}, \tag{15}$$

which can also be written as:

$$(x^1 - y^1)^\top W^{11^\top} W^{11} (x^1 - y^1) \leq \rho_1 (x - y)^T (x - y). \tag{16}$$

Recall that $g(x) = W^{11} x^1 + b^1$ and $g(y) = W^{11} y^1 + b^1$, then the inequality 16 can be written as:

$$||g(x) - g(y)||_2 \leq \sqrt{\rho_1} ||x - y||_2 \quad \text{for all } x, y \in \mathbb{R}^n. \tag{17}$$

Similarly, we have:

$$||h(x) - h(y)||_2 \leq \sqrt{\rho_2} ||x - y||_2 \quad \text{for all } x, y \in \mathbb{R}^n. \tag{18}$$

Given $f(x) = g(x) + h(x)$ in Eq. 12, we can derive:

$$\begin{aligned}
||f(x) - f(y)||_2^2 &= ||(g(x) - g(y)) + (h(x) - h(y))||_2^2 \\
&= ||g(x) - g(y)||_2^2 + ||h(x) - h(y)||_2^2 + 2 \left( g(x) - g(y) \right)^\top \left( h(x) - h(y) \right) \\
&\leq ||g(x) - g(y)||_2^2 + ||h(x) - h(y)||_2^2 + 2||g(x) - g(y)||_2 ||h(x) - h(y)||_2 \\
&\leq \rho_1 ||x - y||_2^2 + \rho_2 ||x - y||_2^2 + 2\sqrt{\rho_1 \rho_2} ||x - y||_2^2 \\
&= (\sqrt{\rho_1} + \sqrt{\rho_2})^2 ||x - y||_2^2.
\end{aligned} \tag{19}$$

Finally, the above inequality implies

$$||f(x) - f(y)||_2 \leq (\sqrt{\rho_1} + \sqrt{\rho_2}) ||x - y||_2 \quad \text{for all } x, y \in \mathbb{R}^n, \tag{20}$$

which gives the upper bound of $L(f) = \sqrt{\rho_1} + \sqrt{\rho_2}$ on the Lipschitz constant of $f(\cdot)$ based on the Definition 1.

## A.2 OBJECT DETECTION ON COCO

In order to better verify the effectiveness of the proposed IMLP module, we conduct experiments for object detection on the COCO 2017 dataset, which contains 118K training images, 5K validation images and 20K test-dev images. Mask R-CNN (He et al., 2017) is considered the object detection framework and Swin-Ti is used as the baseline model. Other training settings are the same as Swin-Ti.

Table 6: Results on COCO object detection.

| Backbone | $AP^{box}$ | $AP_{50}^{box}$ | $AP_{75}^{box}$ | #param | FLOPs |
|---|---|---|---|---|---|
| Swin-Ti | 46.0 | 67.1 | 50.3 | 48M | 267G |
| Swin-Ti + IMLP | 46.0 | 67.2 | 50.3 | 44M | 251G |

We can see in Tab. 6 that our IMLP module can reduce over 4M parameters and 16G FLOPs compared to the original Swin-Ti model with a same box AP, which shows the priority of the proposed method.

## A.3 SEMANTIC SEGMENTATION ON ADE20K

We also conduct experiments for the semantic segmentation task on the ADE20K dataset, which contains 20K training images, 2K validation images and 3K test images from 150 different semantic categories. As in Swin (Liu et al., 2021), we use UperNet (Xiao et al., 2018) as the base semantic segmentation framework and Swin-Ti as the baseline model. Other training settings are the same as Swin-Ti.

Table 7: Results on ADE20K semantic segmentation.

| Backbone | mIoU | mAcc | #param | FLOPs |
|---|---|---|---|---|
| Swin-Ti | 44.5 | 55.6 | 60M | 945G |
| Swin-Ti + IMLP | 45.0 | 57.3 | 56M | 928G |

As shown in Tab. 9, we achieve a 0.5 mIoU improvement while reducing FLOPs by 17G and parameters by 4M compared to the baseline model Swin-Ti.

## A.4 EXPERIMENTS USING DIFFERENT ACTIVATION FUNCTIONS

In the following table, we change the activation function in DeiT from GeLU to SoftPlus, and apply our method using ASoftPlus. The results show that we can achieve consistent improvement regardless of the usage of basic activation functions.

Table 8: Using different activation functions on DeiT-Ti.

| Model | FLOPs (G) | Parameters (M) | Top-1 Acc (%) |
|---|---|---|---|
| DeiT with SoftPlus | 5.72 | 1.26 | 71.6 |
| + IMLP | 5.00 (-12.6%) | 1.10 (-12.7%) | 72.0 |

## A.5 THROUGHPUTS OF THE MODEL

The throughputs with batchsize=32 are shown in the following table. We conduct experiments on the DeiT model. We can see that although the latency improvements on GPU are marginal, the improvements on CPU are obvious and they come from the decrease of FLOPs, which shows the priority of our method. We use V100 GPU and Intel 6136 CPU @ 3GHz CPU in the experiments.

Table 9: Throughputs of DeiT model on CPU and GPU.

| Model | Throughputs on CPU (imgs/s) | Throughputs on GPU (imgs/s) |
|---|---|---|
| DeiT-Ti | 63.6 | 988.6 |
| DeiT-Ti+IMLP | 69.9 (+9.9%) | 1003.6 (+1.5%) |
| DeiT-S | 40.0 | 596.6 |
| DeiT-S+IMLP | 42.9 (+7.3%) | 604.3 (+1.3%) |
| DeiT-B | 20.3 | 275.6 |
| DeiT-B+IMLP | 22.1 (+8.9%) | 277.8 (+0.8%) |

