# OpenReview forum: "Improving MLP Module in Vision Transformer"
_ICLR.cc/2024/Conference — Submitted to ICLR 2024_

### Official Review · Reviewer_JSwb · 2023-10-28

**Soundness:** 3 good
**Presentation:** 4 excellent
**Contribution:** 3 good
**Rating:** 5
**Confidence:** 4

**Summary:**

This paper presents significant enhancements to the design of the MLP module, referred to as IMLP, aimed at augmenting its non-linear capabilities. A key innovation in this work is the introduction of a novel activation function, termed AGeLU. Additionally, a convolution layer has been meticulously crafted to bolster the spatial information within the module. Extensive experimental results are provided in the paper by applying the IMLP module in diverse vision transformer architectures. The results demonstrate that the proposed method can help the transformers obtain similar performance with fewer parameters by reducing the hidden channels of MLP modules.

**Strengths:**

1.	The MLP module's non-linearity is intuitively illustrated in a clear and intriguing manner.
2.	The effectiveness of the proposed method is validated across various tasks and diverse transformer models, encompassing both isotropic and stage-wise variants.
3.	The paper provides a theoretical analysis of the bounds of the modified IMLP module, providing valuable insights for parameter selection within the module.

**Weaknesses:**

1.	The AGeLU's standalone performance appears suboptimal. Table 2 suggests that the primary performance improvements stem from knowledge distillation or spatial enhancement, with the paper lacking a clear demonstration of the enhanced non-linearity's effectiveness, i.e., the proposed AGeLU.
2.	There is ambiguity regarding whether the kernel size is consistently set to 5 for large-scale models such as DeiT-B or Swin-B, lacking a definitive explanation in the paper.
3.	The remarkable performance drops attributed to the addition of GeLU with four times the number of channels raise questions. Since the fully connected (fc) layer entails linear calculations, it appears that the addition operation merely doubles the original output, which requires further clarification.

**Questions:**

Please see the weakness part. Additionally, introducing spatial enhancement after the GeLU operation (as 'a' in Figure 4) would help to conclusively demonstrate the impact of enhanced non-linearity in parameter reduction.

---

> ### Author Response · Authors · 2023-11-15
> **Rebuttal to reviewer JSwb**
>
> **Q1: The AGeLU's standalone performance appears suboptimal. Table 2 suggests that the primary performance improvements stem from knowledge distillation or spatial enhancement, with the paper lacking a clear demonstration of the enhanced non-linearity's effectiveness, i.e., the proposed AGeLU.**
>
> **A1:** There are some misunderstandings and we are sorry for the unclarity.
>
> Firstly, **we do not use KD in our proposed method**. As shown in the first paragraph of section 4.3, using KD method can enhance the performance but will also increase the GPU memory during the training process, which is not feasible to use when the size of the model grows larger. Thus, we propose our spatial-enhancement part in section 4.3. The KD result in Table 2 is only proposed for the completeness of the experiment. This is explained more clearly in Table 2 of the revised version.
>
> Secondly, the usefulness of the channel-wise enhancement part is shown in Table 2 and Table 3. The performance gains indeed come from depthwise convolution (spatial-wise enhancement part). However, **the proposed AGeLU+concat (channel-wise enhancement part) copes very well with dwconv** and can reduce FLOPs and parameters while maintaining accuracy to the greatest extent possible. For example, as shown in Table 2, DeiT-Ti with dwconv only increases 0.6% Top1 accuracy compared to original DeiT-Ti model (compare Line 1 and Line 4). However, by adding AGeLU+concat, DeiT-Ti with dwconv can increase 2.1% Top1 accuracy (compare Line 2 and Line 5) while saving over 10% FLOPs and parameters to the original model (Line 1). Thus, **the proposed spatial and channel enhancement part should be treated as a whole** to enhance the nonlinearity from two different aspects, and achieve the goal of reducing computational cost while maintaining accuracy.
>
> For Table 3, we ablate using the original MLP module (Line 1) and its AGeLU counterpart (Line 2). We also give **the proposed IMLP module with AGeLU+concat (Line 4)** and its GeLU counterpart (Line 3). The result shows that our method of using AGeLU+concat is very useful, and there are some explanations at the end of page 7. We add more explanations below. Line 1 and Line 2 have similar performance since they all have the same degree of nonlinearity according to Eq.2 (the form of original MLP), no matter $\phi=GeLU$ or $\phi$=AGeLU. The reason why Line 4 is much better than Line 3 comes from Eq.6. When using two different AGeLUs, we have different $\alpha_{c’}$ and $\alpha_{c’}'$ to form $t_1$ and $t_1’$ in Eq.6. Thus, the output can be expressed as the linear combination of $C’$ different nonlinear functions. However, when using two GeLU functions, we have $\alpha_{c’}=\alpha’_{c’}=1$. Thus, the output can only be expressed as the linear combination of $C’/2$ different nonlinear functions.
>
> **Q2: There is ambiguity regarding whether the kernel size is consistently set to 5 for large-scale models such as DeiT-B or Swin-B, lacking a definitive explanation in the paper.**
>
> **A2:** Sorry for the unclarity. In fact, we found that the ratio of the computational cost of the MLP module will increase as the scale of the model becomes larger, thus replacing the original MLP module the IMLP module will reduce more FLOPs and parameters and it is harder to make up for the accuracy drop with kernel size=3. Therefore, we consistently set kernel size to 5 for large-scale models.
>
> **Q3: The remarkable performance drops attributed to the addition of GeLU with four times the number of channels raise questions. Since the fully connected (fc) layer entails linear calculations, it appears that the addition operation merely doubles the original output, which requires further clarification.**
>
> **A3:** I believe you are talking about the results shown in Table 4. In fact, the experiments in Table 4 use the following equation to replace the original MLP module, i.e., $y=AGeLU_1(x)+AGeLU_2(x)+\cdot\cdot\cdot+AGeLU_{4C}(x)$. The reason we conduct this ablation study 4 is shown at the beginning of section 4.1. Table 4 is a straightforward way to use the combination of $C^′=4C$ different nonlinear functions to replace the original MLP module. However, based on Corollary 1 (3) in the paper, the bias that depends on the input elements makes it challenging to attain a comparable degree of non-linearity by merely combining multiple nonlinear functions, and the classification performance does not match that of using the original MLP module. This is more clearly explained in the second last paragraph of section 5.2 in the revised version.
>
> **Q4: Introducing spatial enhancement after the GeLU operation (as 'a' in Figure 4) would help to conclusively demonstrate the impact of enhanced non-linearity in parameter reduction.**
>
> **A4:** If my understanding is correct, the result of using spatial enhancement after the GeLU operation is shown in Line 4 of Table 2 (w/ spatial).

---

> ### Author Response · Authors · 2023-11-20
> **Are there any additional questions?**
>
> Dear reviewer JSwb,
>
> As the deadline for the discussion phase is approaching, I would like to inquire if there is anything in my rebuttal that I may not have clarified clearly or if you have any additional questions. I am looking forward to further discuss with you.

---

### Official Review · Reviewer_oReY · 2023-10-30

**Soundness:** 4 excellent
**Presentation:** 3 good
**Contribution:** 4 excellent
**Rating:** 8
**Confidence:** 5

**Summary:**

This paper conducts a theoretical analysis of the MLP module within the architecture of vision transformers, showing that the MLP fundamentally acts as a non-linearity generator. Consequently, the paper proposes an Improved Multilayer Perceptron (IMLP) module, which augments non-linearity across both the channel and spatial dimensions, while concurrently reducing computational complexity by diminishing the hidden dimensions. Experiments suggest that for state-of-the-art models, such as ViT, Swin, and PoolFormer, the substitution of the original MLP with the IMLP module can significantly reduce model complexity without compromising accuracy.

**Strengths:**

1. The paper proposes a solid theoretical analysis by delving into the math of the MLP module, successfully establishing it as a non-linearity generator.
2. This paper introduces AgeLU to form a more nonlinear function and improves the non-linearity within the channel dimension of the IMLP module. The paper extends the non-linearity enhancement to the spatial dimension as well, using a comprehensive approach to improve the IMLP module's capabilities.
3. The empirical validation is compelling, with a variety of architectures and tasks being employed to verify the effectiveness of the proposed IMLP module.
4. The writing in the paper is well done, with a clear structure that makes it easy to understand. The way the ideas are presented is thoughtful and makes for an engaging and informative read.

**Weaknesses:**

1.	In Equation (5), AGeLU and AGeLU′ are introduced as two nonlinear functions. It prompts an intriguing inquiry: what would the outcome be if the division was into more parts, say four? A more comprehensive ablation study should be conducted to provide a richer understanding of the behavior and performance of these functions.
2.	In Section 4.3, there lack of comparative experiments with other non-linear blocks like bottleneck in ResNet or linear bottleneck in MobileNetV2, which could have showcased the unique advantages or potential shortcomings of the proposed method in a broader context.
3.	The proposed IMLP module has only been experimented with a few models like ViT and Swin, which have been proposed for several years. It raises the question of the module's effectiveness on more recent, higher-accuracy models like iFormer. The validation of the IMLP module across a broader spectrum of models could provide a clearer picture of its versatility and efficacy in current vision transformer landscapes.

**Questions:**

See weaknesses above.

---

> ### Author Response · Authors · 2023-11-15
> **Rebuttal to reviewer oReY**
>
> Thanks for the reviewer’s constructive comments and support.
>
> **Q1: In Equation (5), AGeLU and AGeLU′ are introduced as two nonlinear functions. It prompts an intriguing inquiry: what would the outcome be if the division was into more parts, say four? A more comprehensive ablation study should be conducted to provide a richer understanding of the behavior and performance of these functions.**
>
> **A1:** This is a really good question. By dividing the MLP module into more parts, the computational complexity of the model can be reduced, while the degree of non-linearity remains the same which can be derived by simply extending Eq.6 in the main paper. However, as shown at the beginning of section 4.3, the degree of freedom is related to the number of channels in each part, and dividing the MLP module into more parts will decrease the degree of freedom and thus make it hard to achieve a good result. We conduct experiments using two and four nonlinear functions and the results are shown below. We can see that using four nonlinear functions gives a worse classification accuracy compared to using two nonlinear functions, which is consistent with our analysis.
>
> |Model|	FLOPs|	Params|	acc|
> |-|-|-|-|
> |2 nonlinear functions|	1.10|	5.00|	72.6|
> |4 nonlinear functions|	1.01|	4.55|	68.6|
>
> **Q2: In Section 4.3, there lack of comparative experiments with other non-linear blocks like bottleneck in ResNet or linear bottleneck in MobileNetV2, which could have showcased the unique advantages or potential shortcomings of the proposed method in a broader context.**
>
> **A2:** Note that in the original MLP module in different vision transformers, the number of hidden dimensions is four times the input dimensions, which is similar to the architecture of an inverted bottleneck. This is the reason we use the inverted bottleneck in section 4.3. The linear bottleneck has the same hidden dimensions and input dimensions, and the original bottleneck in ResNet has fewer hidden dimensions than the input dimensions, which are not our first choices. The results of different blocks are shown below, and we can see that using an inverted bottleneck gives the best result. In the experiments, as the linear bottleneck and bottleneck will reduce the hidden dimensions, we increase the embedding dimension of the networks to make the FLOPs of different models roughly the same for a fair comparison.
>
> |Model|	FLOPs|	Params|	acc|
> |-|-|-|-|
> |Inverted bottleneck|	1.10	|5.00|	72.6|
> |Linear bottleneck|	1.14|	4.97|	69.9|
> |Bottleneck|	1.11|	4.72|	62.2|
>
> **Q3: The proposed IMLP module has only been experimented with a few models like ViT and Swin, which have been proposed for several years. It raises the question of the module's effectiveness on more recent, higher-accuracy models like iFormer. The validation of the IMLP module across a broader spectrum of models could provide a clearer picture of its versatility and efficacy in current vision transformer landscapes.**
>
> **A3:** We conduct experiments not only on classic models that have been proposed for several years such as ViT and Swin, but also on the latest published popular models such as PoolFormer and portable models LVT which are all published in 2022. We conduct experiments on iFormer-S (NeurIPS 2022) model and the results are shown below. It shows that our method can reduce FLOPs and parameters while keeping the classification performance on iFormer.
>
> |Model|	FLOPs|	Params	|acc|
> |-|-|-|-|
> |iFormer-S|	4.848|	19.866|83.4|
> |iFormer-S + IMLP|	4.330 (-10.7%)|	17.528 (-11.8%)|	83.3|

---

> > ### Comment · Reviewer_oReY · 2023-11-20
> >
> > Thanks for the response from the author, and the rebuttal adequately addresses my concerns. I have also read the comments from other reviewers and observed that the primary points of disagreement revolve around the KD method and the effectiveness and efficiency of DWConv. The author has provided convincing explanations for these aspects. I lean to keep my rating for now.

---

> > > ### Author Response · Authors · 2023-11-22
> > > **Thanks for the reply.**
> > >
> > > Thanks for the support and the efforts you made to review this paper.

---

### Official Review · Reviewer_5FNj · 2023-10-30

**Soundness:** 2 fair
**Presentation:** 2 fair
**Contribution:** 1 poor
**Rating:** 3
**Confidence:** 5

**Summary:**

The authors propose a modified MLP module for vision transformers which involves the usage of the arbitrary GeLU (AGeLU) function and integrating multiple instances of it to augment non-linearity so that the number of hidden dimensions can be reduced. Besides, a spatial enhancement part is involved to further enrich the nonlinearity in the proposed  modified MLP module.

**Strengths:**

1. Base model design is an important topic in our community.
2. This paper is easy to understand.

**Weaknesses:**

1. My major concern is the effectiveness, we can see some parameter and computational cost savings from Table 1, but this method introduced lots of hardware unfriendly operations like DW conv.
2. No actual latencies are provided for the proposed models and the throughput is more critical than the number of parameters and FLOPs for real applications.
3. From Table 3, I can't see a solid improvement from AGeLU.

**Questions:**

Can you offer the actual latencies?

---

> ### Author Response · Authors · 2023-11-15
> **Rebuttal to reviewer 5FNj**
>
> Thanks for your contrastive comments.
>
> **Q1: My major concern is the effectiveness, we can see some parameter and computational cost savings from Table 1, but this method introduced lots of hardware unfriendly operations like DW conv.**
>
> **A1:** DW conv operation is **GPU unfriendly rather than hardware unfriendly**, since the optimization of matrix multiplication on GPU is very well and DW conv has insufficient parallelism on the GPU, not fully utilizing the computational ability. However, this drawback can be largely eased on other computing devices such as CPU, NPU and FPGA, and the advantage of IMLP with fewer FLOPs and parameters can appear. In fact, DW conv is widely used in mobile devices and there are many papers published to speed up dw conv [1], [2].
>
> [1] High Performance Depthwise and Pointwise Convolutions on Mobile Devices. AAAI, 2020.
>
> [2] Designing efficient accelerator of depthwise separable convolutional neural network on FPGA. JSA, 2019.
>
> **Q2: No actual latencies are provided for the proposed models and the throughput is more critical than the number of parameters and FLOPs for real applications.**
>
> **A2:** The throughputs with batch_size=32 are shown in the following table. We conduct experiments on the DeiT model and the results are added in the appendix of the revised version. Throughputs on other models will be added in the final version. We can see that although the latency improvements on GPU are marginal, **the improvements on CPU are obvious** and they come from the decrease of FLOPs, which shows the priority of our method. We use V100 GPU and Intel 6136 CPU @ 3GHz CPU in the experiments.
>
> |Model|	Throughput (imgs/s) on CPU|	Throughput (imgs/s) on V100|
> |-|-|-|
> |DeiT-Ti|	63.6	|988.6|
> |DeiT-Ti+IMLP|	69.9 (+9.9%)|	1003.6 (+1.5%)|
> |DeiT-S|	40.0|	596.6|
> |DeiT-S+IMLP|	42.9 (+7.3%)|	604.3 (+1.3%)|
> |DeiT-B|	20.3|	275.6|
> |DeiT-B+IMLP|	22.1 (+8.9%)|	277.8 (+0.8%)|
>
> **Q3: From Table 3, I can't see a solid improvement from AGeLU.**
>
> **A3:** Table 3 shows that AGeLU is not useful when using it alone (compare Line 1 and Line 2), and **this is not used in our method**. They are listed here for the completeness of this ablation study. In fact, our proposed channel-wise enhancement part with AGeLU+concat has a non-negligible 0.3% accuracy improvement compared to using GeLU+concat (compare Line 4 and Line 3). Thus, we need to **treat the channel-wise enhancement part as a whole and a solid improvement can be seen**. More explanation of Table 3 can be found at the end of page 7.

---

> ### Author Response · Authors · 2023-11-20
> **Are there any additional questions?**
>
> Dear reviewer 5FNj,
>
> As the deadline for the discussion phase is approaching, I would like to inquire if there is anything in my rebuttal that I may not have clarified clearly or if you have any additional questions. I am looking forward to further discuss with you.

---

> ### Comment · Reviewer_5FNj · 2023-11-22
>
> Thanks for the response. These models (the authors offered in the rebuttal) are not designed for CPUs or any other edge devices, it's more for GPUs. So it's not convincing to say this work is for mobile devices. The gap between throughput (0.8-1.5%) on GPU VS the theory percentage (12.7-15.1%) is too big. I recommend authors work more on mobile models in the future if you want to claim the effectiveness of this work on mobile devices. I will keep my rating.
>
> Best

---

> ### Author Response · Authors · 2023-11-22
> **Thanks for the reply.**
>
> Thanks for the reply from the reviewer.
>
> However, I must point out that this paper is not designed for any specific devices such as GPU, CPU, or other edge devices since both cumbersome and portable ViT models have MLP modules and we focus on improving the MLP modules.
>
> It is also not true that this paper is 'more for GPUs', since we give the experimental results for both large and portable models in Table 1 in the paper.
>
> Anyway, thanks for your effort to review this paper.

---

> > ### Comment · Reviewer_5FNj · 2023-11-22
> >
> > Thanks for the response.
> >
> > From the throughput you offered in rebuttal, the DeiT models are designed for GPUs, but you claimed: "the improvements on CPU are obvious". And I didn't see the throughput evaluation on efficient ViT.
> >
> > Best

---

> ### Author Response · Authors · 2023-11-22
> **Thanks for the reply.**
>
> Thank you for pointing this out. We list the throughput of LVT [1] on CPU and the proposed method in the following table, and hope this can address your concern.
>
> |Model|	Throughput (imgs/s) on CPU|
> |-|-|
> |LVT-R1|	81.5	|
> |LVT-R1+IMLP|	86.8 (+6.5%)|
> |LVT-R2|	79.7|
> |LVT-R2+IMLP|	85.1 (+6.8%)|
> |LVT-R3|	74.5|
> |LVT-R3+IMLP|78.9	 (+5.9%)|
> |LVT-R4|71.6	|
> |LVT-R4+IMLP|	75.3 (+5.2%)|
>
> [1] Lite Vision Transformer with Enhanced Self-Attention.

---

> ### Comment · Reviewer_5FNj · 2023-11-22
>
> Thanks for your effort. As the LVT's authors test their throughput on GPU, you should keep the same. And for efficient ViTs, it's more convincing to test the effectiveness of this work on models such as EfficientViT[1].
>
>
>
> [1] Liu, Xinyu, Houwen Peng, Ningxin Zheng, Yuqing Yang, Han Hu, and Yixuan Yuan. "EfficientViT: Memory Efficient Vision Transformer with Cascaded Group Attention." In Proceedings of the IEEE/CVF Conference on Computer Vision and Pattern Recognition, pp. 14420-14430. 2023.

---

> ### Author Response · Authors · 2023-11-22
> **Thanks for the reply.**
>
> We show the throughputs on GPU and CPU for LVT in the following table. Though our method is slightly slow on GPU, it is much faster on CPU.
>
> |Model|	Throughput (imgs/s) on CPU|	Throughput (imgs/s) on GPU|
> |-|-|-|
> |LVT-R1|	81.5	|1253.4|
> |LVT-R1+IMLP|	 (+6.5%)|1241.6 (-0.9%)
> |LVT-R2|	79.7|	1131.0|
> |LVT-R2+IMLP|	85.1 (+6.8%)|1112.6 (-1.6%)
> |LVT-R3|	74.5|	1034.1|
> |LVT-R3+IMLP|78.9	 (+5.9%)|1015.9 (-1.7%)|
> |LVT-R4|71.6	|	943.2|
> |LVT-R4+IMLP|	75.3 (+5.2%)| 926.3 (-1.7%)|
>
> For the comparison with EfficientViT, since the deadline of the discussion phase is about to end, it is hard for us to implement the code, train the model, and test the classification performance and throughput. But we will cite EfficientViT and add the comparison in the final version.
>
> Thank you.

---

### Official Review · Reviewer_as7x · 2023-11-02

**Soundness:** 2 fair
**Presentation:** 2 fair
**Contribution:** 2 fair
**Rating:** 3
**Confidence:** 5

**Summary:**

This paper introduces a derivative of GeLU called arbitrary GeLU (AGELU) that aims to improve the capability of MLP in vision transformers. AGeLU is used in the MLP and is further combined with the concatenation operators, and extra spatial depthwise convolution (DWConv) following the tuple of BN/GELU  is included in the end. The authors provide some theoretical backups for justifying the proposed element with the experiments on ImageNet-1K.

**Strengths:**

- The idea is simple and easily applicable to any vision transformers.
- Some theoretical justifications are provided.

**Weaknesses:**

- This paper primarily addresses the activation functions, but many related works are missing, which have emerged after GeLU:
  - Mish: A Self Regularized Non-Monotonic Activation Function, BMVC 2020
  - Padé Activation Units: End-to-end Learning of Flexible Activation Functions in Deep Networks, ICLR 2020
  - Smooth Maximum Unit: Smooth Activation Function for Deep Networks using Smoothing Maximum Technique, CVPR 2022

- The performance enhancements provided by the proposed activation function are marginal and non-existent in some instances. The proposed activation function fails to improve accuracy in larger models like Deit_B and LVT-R4; moreover, it actually leads to a decline in performance for Swin-B and Poolformer-M48.

- The main issue identified by the reviewer is the performance gains of this work appear to depend largely on employing depthwise convolution, which has been already recognized in many prior hybrid architectures. The ablation studies presented in the manuscript further underscore this reliance as well. As a result, the paper's contribution is considered to be very limited.

- This paper needs more experiments to justify the claim:
  - Experimental comparisons with simple simple activation functions such as SoftPlus, ELU, ReLU6, Swish, and so on are not compared.
  - Downstream tasks in the Appendix contain limited results with a few backbones.

- It is speculated that the proposed method's effectiveness relies on KD (Table 2 evidently shows this), which requires a teacher model. Consequently, training budgets may not be preserved equally.

- The reviewer acknowledges that while the theories included do enhance the paper, it lacks a crucial explanation—specifically, the rationale behind why AGeLU with concatenation is necessary has not been addressed via theory.

- The proposed variant of GeLU is not exclusively applicable to vision transformers. It can also be utilized in architectures like ConvNeXt, which shares similar building blocks, excluding self-attention, where the proposed element could serve as a replacement for standard GeLUs.

**Questions:**

See above weaknesses.

- Please specify how KD works when training with the proposed activation function.

- The reviewer highlights Table 3, noting it presents a surprising and crucial result of the study. The authors are requested to provide insights or intuitions into why such an outcome occurred.

- The results in Table 4 are not clearly explained.

- Why is the additional shortcut needed for the dwconv and BN should follow it subsequently?

Pre-rebuttal comments) This paper proposes a variant of GeLU to improve the MLP module in the vision transformer module. However, the identified shortcomings and the raised questions lead to the conclusion that the paper does not meet the publication standards of ICLR in its present state. I would like to see the authors' responses and the other reviewer's comments.

---

> ### Author Response · Authors · 2023-11-15
> **Rebuttal to reviewer as7x (part 1/2)**
>
> Thanks for your contrastive comments.
>
> **Q1: This paper primarily addresses the activation functions, but many related works are missing, which have emerged after GeLU.**
>
> **A1:** This is a good question and we will add the corresponding related works. **Note that we actually propose a general arbitrary nonlinear function in Eq.3 rather than a specific AGeLU**, and the basic function $\phi$ in Eq.3 is not restricted to the GeLU function. As shown in the last sentence in the paragraph below Eq.4, other basic activation functions can be extended using the same way. The reason we chose GeLU as the basic function is because the activation functions used in the baseline models are GeLU. In the following table, we change the activation function in DeiT from GeLU to SoftPlus, and apply our method using ASoftPlus. The results show that we can achieve consistent improvement regardless of the usage of basic activation functions (The experimental results are added in the appendix of the revised version). Other activation functions such as ELU, ReLU6 and Swish will be added in the final version.
> |Model|FLOPs|Params|acc|
> |-|-|-|-|
> |DeiT with SoftPlus|	5.72|1.26|71.6|
> |+ IMLP|	5.00 (-12.6%)|	1.10 (-12.7%)|	72.0|
>
> **Q2: The performance enhancements provided by the proposed activation function are marginal and non-existent in some instances. The proposed activation function fails to improve accuracy in larger models like Deit_B and LVT-R4; moreover, it actually leads to a decline in performance for Swin-B and Poolformer-M48.**
>
> **A2:** The proposed IMLP aims to reduce the computational cost while maintaining the classification performance compared to the baseline model, rather than improve the accuracy only. As shown in **Figure 1**, the proposed method has **a better FLOPs-Accuracy trade-off** among all the baseline methods. This is because the FLOPs can reduce significantly with a slight or no performance drop.
>
> **Q3: The main issue identified by the reviewer is the performance gains of this work appear to depend largely on employing depthwise convolution, which has been already recognized in many prior hybrid architectures. The ablation studies presented in the manuscript further underscore this reliance as well. As a result, the paper's contribution is considered to be very limited.**
>
> **A3:** This is also a very good question. The performance gains indeed come from depthwise convolution (spatial-wise enhancement part). However, **the proposed AGeLU+concat (channel-wise enhancement part) copes very well with dwconv** and can reduce FLOPs and parameters while maintaining accuracy to the greatest extent possible. For example, as shown in Table 2, DeiT-Ti with dwconv only increases 0.6% Top1 accuracy compared to original DeiT-Ti model (compare Line 1 and Line 4). However, by adding AGeLU+concat, DeiT-Ti with dwconv can increase 2.1% Top1 accuracy (compare Line 2 and Line 5) while saving over 10% FLOPs and parameters to the original model (Line 1). Thus, the proposed **spatial and channel enhancement part should be treated as a whole** to enhance the nonlinearity from two different aspects, and achieve the goal of reducing computational cost while maintaining accuracy.
>
> Besides, although many papers have used dwconv in their transformer, they do not give a thorough analysis of the reason it works (which is also mentioned at the end of page 1). In this paper, we treated **dwconv as the enhancement of nonlinearity** which explains the usefulness of dwconv in a new perspective.
>
> **Q4: This paper needs more experiments to justify the claim: Experimental comparisons with simple activation functions such as SoftPlus, ELU, ReLU6, Swish, and so on are not compared. Downstream tasks in the Appendix contain limited results with a few backbones.**
>
> **A4:** Thanks for the suggestion. The results of using SoftPlus are shown in the answer of Q1. Other activation functions and experiments on more downstream tasks will be added in the final version due to the limited rebuttal period.
>
> **Q5: It is speculated that the proposed method's effectiveness relies on KD (Table 2 evidently shows this), which requires a teacher model. Consequently, training budgets may not be preserved equally.**
>
> A5: This is a misunderstanding and we are sorry for the unclarity. **We do not use KD in our proposed method**. As shown in the first paragraph of section 4.3, using KD method can enhance the performance but will also increase the GPU memory during the training process, which is not feasible to use when the size of the model grows larger. Thus, we propose our spatial-enhancement part in section 4.3. The KD result in Table 2 is only proposed for the completeness of the experiment. This is explained more clearly in Table 2 and the corresponding explanation of the revised version.

---

> ### Author Response · Authors · 2023-11-15
> **Rebuttal to reviewer as7x (part 2/2)**
>
> **Q6: The reviewer acknowledges that while the theories included do enhance the paper, it lacks a crucial explanation—specifically, the rationale behind why AGeLU with concatenation is necessary has not been addressed via theory.**
>
> **A6:** The reason for using AGeLU with concatenation is shown in Eq.2 (the form of original MLP) and Eq.6 (the form of MLP using AGeLU with concatenation) and also in the last paragraph of section 4.1. I will try to explain it more clearly in the following.
> We can rewrite the last line of Eq.6 into the form:
>
> $\mathbf y'=\mathbf t_2 \mathbf W^e=\left(\sum_{j=1}^{C'}w_{jc}^e\cdot[\beta_{j}{\rm{GeLU}}(\alpha_{j}\sum_{i=1}^{C}w_{i,f(j)}^dx_i+\gamma_{j})+\theta_{j}] \right)_{c=1}^C$
>
> $~~~~~~~~~~~~~~~~~~~=\left(\sum_{j=1}^{C'}w_{jc}^{'e}{\rm{GeLU}}(m_{cj}'x_c+n_{cj}')+\theta_{j}\right)_{c=1}^C,~~~~~~~~Eq.6'$
>
> where $w_{jc}^{'e}=w_{jc}^{e}\cdot\beta_j$, $m_{cj}'=w^d_{c,f(j)}$ and $n_{cj}'={\rm{func}}(x_1,\cdot,\cdot,\cdot,x_{c-1},x_{c+1},\cdot,\cdot,\cdot,x_C)=\sum_{i=1,i\neq c}^C{w_{i,f(j)}^dx_i+\gamma_{j}}$.
>
> This is almost the same as Eq.2. According to Corollary 1, the output of Eq.2 and Eq.6’ can both be expressed as the combination of C’ different nonlinear functions with distinct scales and biases, in which the scales are learnable weights independent to the input and the biases are dependent to the input.
>
> The only difference between Eq.2 and Eq.6’ is that the degree of freedom of  $w_{jc}^{'e}$ in Eq.6’ is halved compared to the original $w_{ij}^a$ in Eq.2. Thus, it is harder to optimize Eq.6’ and may degrade the performance, as shown in the first paragraph in section 4.3.
>
> The above analysis is added near Eq.6 in the revised version.
>
> **Q7: The proposed variant of GeLU is not exclusively applicable to vision transformers. It can also be utilized in architectures like ConvNeXt, which shares similar building blocks, excluding self-attention, where the proposed element could serve as a replacement for standard GeLUs.**
>
> **A7:** This is true. However, the analysis in this paper is based on a 2-layer MLP module and thus is only applicable to vision transformers. Using it to ConvNeXt and other MLP networks is a really good topic and future work that is worth discussing.
>
> **Q8: Please specify how KD works when training with the proposed activation function.**
>
> **A8:** **We do not use KD in our proposed method.** The KD result in Table 2 is only proposed for the completeness of the experiment (please see A5 for more explanation). In detail, we transfer the knowledge from the output of each original MLP module in the teacher model to the output of each proposed IMLP module in the student model. Also, the classification probabilities are also distilled.
>
> **Q9: The reviewer highlights Table 3, noting it presents a surprising and crucial result of the study. The authors are requested to provide insights or intuitions into why such an outcome occurred.**
>
> **A9:** Some of the explanations are shown at the end of page 7. In fact, Line 1 of Table 3 stands for the original MLP module, and Line 2 changes the GELU function to AGeLU. These two architectures have similar performance since they all have the same degree of nonlinearity according to Eq.2 (the form of original MLP), no matter $\phi$=GeLU or $\phi$=AGeLU.
>
> Line 4 of Table 3 is the proposed IMLP module using 2 different AGeLU functions with concat, and Line 3 uses two GeLU functions with concat. The reason why Line 4 is much better than Line 3 comes from Eq.6. When using two different AGeLUs in Line 4, we have different $\alpha_{c’}$ and $\alpha_{c’}'$ to form $t_1$ and $t_1’$ in Eq.6 in the original paper. Thus, the output can be expressed as the linear combination of $C’$ different nonlinear functions. However, when using two GeLU functions in Line 3, we have $\alpha_{c’}=\alpha_{c’}'=1$. Thus, the output can only be expressed as the linear combination of $C’/2$ different nonlinear functions.
>
> **Q10: The results in Table 4 are not clearly explained.**
>
> **A10:** Sorry for the unclarity. In fact, the explanation of Table 4 is shown at the beginning of section 4.1. Table 4 is a straightforward way to use the combination of C′ different nonlinear functions to replace the original MLP module. However, based on Corollary 1 (3) in the paper, the bias that depends on the input elements makes it challenging to attain a comparable degree of non-linearity by merely combining multiple nonlinear functions, and the classification performance does not match that of using the original MLP module.
>
> **Q11: Why is the additional shortcut needed for the dwconv and BN should follow it subsequently?**
>
> **A11:**  Many researchers found that learning the residual of the output is easier than learning the original input, and BN can normalize the input for better optimization. We found that using BN and shortcut can achieve a better result than not using it through empirical experiments.

---

> ### Author Response · Authors · 2023-11-20
> **Are there any additional questions?**
>
> Dear reviewer as7x,
>
> As the deadline for the discussion phase is approaching, I would like to inquire if there is anything in my rebuttal that I may not have clarified clearly or if you have any additional questions. I am looking forward to further discuss with you.

---

> > ### Comment · Reviewer_as7x · 2023-11-22
> > **Official Comment by Reviewer as7x**
> >
> > Thank you for the responses to my concerns. Some of my concerns still remain:
> > - The authors argue that their method combined with depthwise convolution (dwconv) leads to a significant improvement of +2.1pp from a 70.5 baseline, compared to dwconv alone, which offers only a +0.6pp improvement from a 72.2 baseline. However, this comparison is questionable because the 70.5 baseline itself is lower due to using a less effective GELU variant. Table 3 indicates that AGeLU alone does not beat GELU, and while AGeLU with concatenation does offer some improvement (+0.4pp), it's still less than the +0.6pp by dwconv. This suggests that dwconv contributes more to the overall improvement. (plus, there's a minor concern about whether the comparison in Table 3, particularly in terms of the number of parameters and FLOPS, is fair).
> >
> > - I was curious about the need for an additional shortcut with BN, given that there's already a shortcut outside the MLP block, which would make the extra shortcut redundant.
> >
> > Currently, while the authors have addressed many concerns, key experiments and materials should be revised, crucial for strengthening the manuscript might not be feasible in this round; I will keep my rating.

---

> ### Author Response · Authors · 2023-11-22
> **Thanks for your reply.**
>
> **Answer to new Q1:**
>
> We should explain Table 2 in this way. In the first line, the original DeiT-Ti model has 72.2% accuracy. After adding the AGeLU+Concat, the performance on line 2 decreased to 70.5% while at the same time has less FLOPs and parameters. Then, we add dwconv and the performance increases +2.1pp and reaches 72.6% at line 4. The +2.1pp improvement is much more significant than adding dwconv alone to the original DeiT-Ti model which has +0.6pp improvement at line 3. Thus, gathering AGeLU+Concat with dwconv is particularly useful and should be treated as a whole.
>
> Overall, using the channel-enhancement part alone decreases the performance while at the same time reducing FLOPs and parameters, using the spatial-enhancement part alone increases the performance but also increases FLOPs and parameters. Gathering both of them can get a better accuracy-FLOPs trade-off.
>
> In Table 3, Line 1 and Line 2 has the same FLOPs and parameters, and Line 3 and Line 4 have the same FLOPs and parameters. Thus, the comparison between GeLU & AGeLU and GeLU+concat & AGeLU+concat are fair.
>
> **Answer to new Q2:**
>
> It is not a redundant shortcut. We can give the equation of IMLP module without using shortcut in the spatial-enhancement part as $y = x + FC(Spatial(z))$, where $z=Channel(FC(x))$. When using shortcut, the equation becomes $y = x + FC(Spatial(z)+z)$. Since there are non-linear activations in $Spatial(\cdot)$ (GeLU function as shown in Figure 4 (b)), the $Spatial(z)+z$ part cannot be merged.
>
> Besides, we found a practical performance improvement by using shortcut and BN.
>
> **Currently, while the authors have addressed many concerns, key experiments and materials should be revised, crucial for strengthening the manuscript might not be feasible in this round.**
>
> **Answer:** Most of the explanations are already shown in the original paper and we only need to reorganize them. Minor experiments can be moved to the appendix to save space for important experiments and materials. These modifications are already updated in the revised paper so that you can check them.

---

### Author Response · Authors · 2023-11-22
**Revised version of the paper is updated.**

Dear reviewers and ACs, you can check the latest version of our paper, in which most of the concerns are included.

Best,

Authors of submission 4940

---

### Meta-Review · Area_Chair_GBXQ · 2023-12-05

**Metareview:**

The paper proposes a modifier MLP design ("Improved MLP", or "IMLP") for vision transformers. Swapping a usual MLP by IMLP allows achieving the same classification performance while using slightly less FLOPs and parameters.

After the rebuttal and discussion, the evaluation of the paper by the reviewers is mostly negative, but not entirely. Below are key pros and cons.

Pros:
1. The proposed module is generally applicable
2. There is some interesting theoretical analysis
3. There are some performance gains from the proposed method

Cons:
1. The baselines are only fairly vanilla vision transformer versions. More have been shown in the rebuttal, but still, if efficiency is of interest, there are many more models to be considered, e.g. EfficientViT that is way more efficient.
2. Comparisons are mostly presented in terms of FLOPs, not actual throughputs. There is now a section in the appendix about throughputs, but those should be much more central.
3. Limited empirical gains - 10-20% decrease in FLOPs and substantially less in actual throughput (on GPU - much less, up to even a slight decrease)

Overall, while the paper is somewhat interesting, in the end it's a fairly small twist on standard architectures, so more thorough baseline comparisons and larger empirical gains would be needed to make the paper broadly interesting and useful. Hence, at this point, I cannot recommend the paper for publishing.

**Justification For Why Not Higher Score:**

it's a fairly small twist on standard architectures, so more thorough baseline comparisons and larger empirical gains would be needed to make the paper broadly interesting and useful

**Justification For Why Not Lower Score:**

N/A

---

### Decision · Program_Chairs · 2024-01-16

Reject